# Endophytes from Halotolerant Plants Aimed to Overcome Salinity and Draught

**DOI:** 10.3390/plants11212992

**Published:** 2022-11-06

**Authors:** Vladimir K. Chebotar, Elena P. Chizhevskaya, Maria E. Baganova, Oksana V. Keleinikova, Oleg S. Yuzikhin, Alexander N. Zaplatkin, Olesya V. Khonina, Roman D. Kostitsin, Nina G. Lapenko

**Affiliations:** 1All-Russia Research Institute for Agricultural Microbiology, Podbelskogo hwy, 3, Pushkin, St. Petersburg 196608, Russia; 2North Caucasus Federal Scientific Agrarian Center, Federal State Budgetary Scientific Institution, Stavropol Territory, Nikonova str., 49, Shpakovsky District, Mikhailovsk 356241, Russia

**Keywords:** endophytes, halophytic plants, *Bacillus* sp., *Salicornia europaea* L., *Salsola australis* (R.Br.), *Kochia prostrata* (L.) Schrad., *Bassia sedoides* (Pall.), plant growth promotion, antifungal activity, bioinoculant

## Abstract

The aim of our research was to study the endosphere of four halophytic plants: *Salicornia europaea* L., *Salsola australis* (R.Br.), *Bassia sedoides* (Pall.) and *Kochia prostrata* (L.) Schrad. from arid and saline areas of the Stavropol Territory, Russia. In total, 28 endophyte strains were isolated from the roots and stems of these halophytic plants. Most of the isolates (23 out of 28) were identified as *Bacillus* sp. while others belonged to the genera *Oceanobacillus, Paenibacillus, Pantoea, Alcaligenes* and *Myroides*. Three strains of *Bacillus* sp. (Se5R, Se1-1R, and Se1-3S), isolated from the *S. europaea* were capable of growth at 55 °C and in 10% of NaCl. Strains Se1-4S, Kp20-2S, and Bs11-2S *Bacillus* sp. (isolated from the *S. australis*, *K. prostrata* and *B. sedoides,* respectively) demonstrated strong plant growth promoting activity: 85–265% over control lettuce plants and a high degree of growth suppression (59.1–81.2%) of pathogenic fungi *Fusarium oxysporum*, *Bipolaris sorokiniana* and *Rhizoctonia solani*. Selected strains can be promising candidates for the development of bioinoculants to facilitate salt soil phytoremediation and be beneficial for mitigating the salt stress to the plants growing in salt-affected habitats.

## 1. Introduction

Environmental stresses such as drought, temperature, salinity, air pollution, heavy metals, pesticides, and soil pH are major limiting factors in crop production because they affect almost all plant functions [1]. The agricultural land is decreasing constantly due to population pressure, adverse environmental conditions, and global climate change [2,3,4].

The total area of salt-affected soils in the world is 831 million hectares [5]. More than 45 million hectares of irrigated land are affected by salt which accounts for 20% of total land and 1.5 million hectares of land are taken out of production each year owing to high salinity levels [6,7]. Salinity is a continuous process and its remediation is cost- and labor-intensive. It is a complex global problem that cannot be solved simply, rather a multidisciplinary approach is required. There are various ways for the remediation and proper utilization of saline soils including agronomic practices, the use of salt-tolerant crop varieties, and phytoremediation. Among them, phytoremediation can be a cost-effective and environmentally sound technology for the remediation of salt-impacted sites [8]. Revegetation of saline soils with halophytes is regarded as a proactive phytoremediation method to improve soils [9,10]. 

In the world, there are about 300,000 species of plants, and each species may be a host for one or more species of endophytic microorganisms and these habitat-adapted symbiotic endophytes can have profound effects on host plant stress tolerance and fitness [11,12,13]. Endophytic bacteria can be used to overcome the effect of salinity stress, and promote plant growth and nutrient uptake; these approaches can provide beneficial and environmentally friendly solutions for sustainable global food security [14,15,16,17]. 

Halophytes could be valuable sources of novel endophytic isolates that can be used to overcome various biotic and abiotic stresses [18,19,20,21]. Endophytes can contribute to plant health impeded by salinity stress, by influencing phytohormonal levels and signaling, contributing to homeostasis maintenance of toxic ions under salinity stress, enhancing photosynthesis, and contributing to biomass production and allocation [22,23]. Besides that, endophytes can have a positive effect on their host plants by fixation of atmospheric nitrogen, solubilizing of soil nutrients, and synthesis of some natural products that protect host plants against many biotic and abiotic factors [24,25].

For example, strains of *Arthrobacter agilis*, *Bacillus endophyticus*, *Bacillus tequilensis*, *Planococcus rifietoensis*, *Variovorax paradoxus*, and ISE-12 *Pseudomonas* sp., isolated from the halophyte *Salicornia europaea* L. were able to facilitate seed germination and seedling growth in the presence of growth inhibitory levels of salt [26,27]. A similar effect was demonstrated by the strain UAE 1 *Micromonospora chelsea* isolated from *Salicornia bigelovii* Torr. [28]. 

The *Glutamicibacter halophytocola* strain KLBMP 5180 isolated from root tissue of halophyte *Limonium sinense* (Girard) Kuntze significantly promoted host growth under NaCl stress [29]. Endophytes of plants from the Tar desert (desert of Pakistan) were studied. It was demonstrated that three strains—PK6-15 *Bacillus cereus* (endophyte of *Zygophyllum simplex* L.), PK5-26 *B. subtilis* (endophyte of *Tribulus terrestris* L.), and PK3-109 *B. circulans* (endophyte of *Panicum antidotale* Lam. Retz.) significantly improved plant growth under salt stress, doubling the weight of fresh plants compared to non-inoculated plants [30]. 

So, the isolation of functionally valuable endophyte partners for halophytic plants, and the selection of compatible and effective plant-bacterial systems for agricultural production on saline soils is a promising scientific direction. 

In our work, we analyzed the endosphere of halophyte plants growing in saline and arid regions of the Stavropol Territory (Russia). Neftekumsky and Levokumsky districts in Stavropol Territory belong to an extremely arid zone and occupy landscapes of semi-desert steppes. The climate of the area is characterized as extremely continental, with an average annual precipitation of 384 mm. In summer, the east wind brings the hot air of the Central Asian deserts. It is associated with droughts and dust storms that begin at a wind speed of 15–20 m/s and the soil temperature can reach 55–60 °C. Halophytic plants in this area are represented by *Salicornia europaea* L., *Salsola australis* (R.Br.), *Bassia sedoides* (Pall.) and *Kochia prostrata* (L.) Schrad. All these plants belong to the family Amaranthaceae. 

*Salicornia europaea* L. is a halophyte, and among all halophyte plants tolerates the highest level of salt (up to 3% NaCl) [31]. In fact, Salicornia name originated from the Latin word meaning “salt” [32]. This plant has spongy stems with diminutive scale-like leaves, and inconspicuous flowers, and fruits. *S. europaea* has been cultivated for soil desalination because it hyperaccumulates salt. This plant is of big ecological importance and commercial value [26]. In addition, the plant is edible and even considered a delicacy, and is also used for medicinal purposes. 

*Salsola australis* (R.Br.) (homotypic synonym *Kali australe* (R.Br.) is an annual drought-resistant plant with a height of 10–80 cm with a root system reaching 2 m. The stem is strongly branched from the base, so the aboveground part has a hemispherical or spherical shape. Young plants of *S. australis* are suitable for silage, and can also be eaten. 

*Bassia sedoides* (Pall.) (homotypic synonym *Grubovia sedoides* (Pall.) is an annual gray-green succulent 10–15 cm high, halophyte, that grows on clay saline areas in the steppes, on salt marshes. The plant is pubescent, the stem is covered with curly hairs. In semi-desert areas, it serves as fodder for camels and small cattle. 

*Kochia prostrata* (L.) Schrad. (homotypic synonym *Bassia prostrata* (L.) A.J.Scott) is a perennial semi-shrub of gray-green color with a height of 30–50 cm with a well-developed root system and is an extremely drought and salt-resistant plant. *K. prostrata* is food for many animals, because it has a high nutritional value. 

The aim of our research was to study the endosphere of these halophytic plants from extreme regions of Stavropol territory, isolate endophytic bacteria from roots and stems of halophytes and study their beneficial properties which will be helpful for the plants to overcome abiotic stresses. In the future these bacteria could be applied as bio-inoculants to facilitate salt soil phytoremediation and be beneficial for mitigating the salt stress to the plants growing in such salt-affected habitats.

## 2. Results

### 2.1. Plant Sampling and Measurement of Salt Content in the Soil

Before collecting the halophytic plants we selected two places where may be possible to find plants of interest growing at the same point near each other. The first place was near the village of Winter Rate Neftekumsky district, Stavropol Territory, 44.5453 N 45.1907 E (Appendix A). The soil of the site was a solonchak (meadow salt marsh). Measurement of salt content in the soil showed electrical conductivity (EC) = 4692 µS/cm or total dissolved solids (TDS) = 3003 ppm. Here were collected plant samples of *Salicornia europaea* L., *Salsola australis* (R.Br.) and *Bassia sedoides* (Pall.) (Appendix A). Totally were collected 15 plant samples (5 samples of each halophyte). The second place was near the village of Burgun-Majary (Kurgan) Levokumsky district, Stavropol Territory, 44.5017 N 44.2901 E. The soil of the site was kastanozem (FAO soil group). Measurement of salt content in the soil showed electrical conductivity (EC) = 57 µS/cm or TDS= 69 ppm. Here were collected plant samples of *Kochia prostrata* (L.) Schrad and *Salsola australis* (R.Br.) (Appendix A). Totally were collected 10 plant samples (5 samples of each halophyte). Samples were collected on 17–18 May 2022 at t 25 °C and humidity at 55%.

### 2.2. Isolation of Endophytic Bacteria

The protocol for isolating endophytes requires surface sterilization of plant tissues, followed by their washing, maceration, sequential dilution, and application of a suspension to a nutrient medium [33]. Sterilizing agents such as sodium hypochlorite, ethanol, and hydrogen peroxide are commonly used for surface sterilization, and usually, these chemicals are used sequentially to increase the effectiveness of the sterilization procedure [34]. Firstly we tried to use sterilization protocol for the external parts of plants studied according to [35]. However, we observed bacterial growth on the nutrient agar plates when spread solutions were obtained after the final washing. Therefore, we developed our own protocol by significantly increasing exposition time in 70% ethanol and in 18% of H_2_O_2_ (see Section 4.2). The resulting sterilization protocol is quite aggressive, but only such sterilization of plant material gave us the opportunity to isolate endophytic strains, and not epiphytic ones.

After surface sterilization of plant material endophytes were isolated from the stems and roots of collected plants (Table 1). A total of 28 isolates were isolated: 13 isolates from halophytes collected in the Neftekumsky district (8 from roots and 5 from stems) and 15 isolates from halophytes collected in the Levokumsky district (8 from roots and 7 from stems). It was not possible to isolate endophytes from some plants. For example, only one strain of endophytic bacteria was isolated from the roots of one of five samples of *Salsola australis* (R.Br.), collected in the Neftekumsky district. Isolated endophytes were stored at a temperature of −80 °C.

### 2.3. Molecular Identification of Endophytes

Most of the isolates (23 out of 28) were identified as *Bacillus* spp. At that, for some isolates, the nucleotide sequences of the 16S rRNA gene were absolutely identical. Therefore, we divided the isolates of the genus *Bacillus* into nine groups depending on the similarity of 16S rDNA (Table 2). Thus, group 1 contains 11 isolates having absolutely identical sequences of the 16S rRNA gene. On the same basis, the isolates Se5R and Bs11-2S were grouped into group 5, the isolates Se1-1R, Se1-3S, and Sa21S were grouped into group 7, and the isolates Sa22-2S and Sa25-2S were grouped into group 9. Each of the remaining isolates of the genus *Bacillus* has had different sequences of the 16S rRNA gene from the rest and is isolated in a separate group (groups 2–4, 6, 8). For each of the groups, representatives with the closest 16S rDNA sequences were identified in the GenBank database (Table 2). The phylogenetic analysis is presented in Figure 1. The remaining 5 isolates were identified as: Se3R—*Oceanobacillus* sp., Kp20-1R as *Paenibacillus* sp., Kp16R as *Pantoea* sp., Sa24-1R as *Alcaligenes* sp. and Sa25-1S—as *Myroides* sp. (Table 2, Figure 1). 

### 2.4. The Effect of Temperature on the Growth of Endophytes

We analyzed the ability of endophytes to grow at temperatures of 4 °C, 10 °C, 15 °C, 20 °C, 28 °C, 37 °C, 45 °C, 55 °C and 60 °C (Table 3). 

All strains were able to grow in the range of 20–37 °C, except the strain Sa25-1S *Myroides* sp. (no growth at 37 °C). Fifteen isolates of *Bacillus* sp., as well as strains Se3R *Oceanobacillus* sp., Kp16R *Pantoea* sp., Sa24-1R *Alcaligenes* sp., and Sa25-1S *Myroides* sp. were capable of growth at 15 °C. At 10 °C, only four isolates of endophytes (Se1-3S, Sa22-2S, Sa25-2S) belonging to *Bacillus* sp. and Kp16R *Pantoea* sp. were able to grow. Growth at 45 °C was observed for all *Bacillus* sp. isolates except strain Sa25-2S, as well as for the strain Kp20-1R *Paenibacillus* sp. Nine isolates of *Bacillus* sp. were capable of growth at 55 °C (Table 3). Of them, five were isolated from *Salicornia europaea* L. (Se1-4S, Se4-2S, Se4-1R, Se1-1R, Se1-3S), two from *Bassia sedoides* (Pall.) (Bs12R, Bs11-1R), one from *Kochia prostrata* (L.) Schrad. (Kp19-3S) and one from *Salsola australis* (R.Br.) (Sa22-1R). At 4 °C and 60 °C, not a single strain was able to grow.

### 2.5. The Effect of Different Concentrations of NaCl on the Growth of Endophytes

We analyzed the ability of the isolates to grow at 1%, 5%, 10%, 15%, and 20% NaCl (Figure 2). All strains were able to grow on a GMF medium with 1% NaCl. All isolates related to *Bacillus* sp., as well as strains Se3R *Oceanobacillus* sp. and Kp16R *Pantoea* sp. grew on a medium with 5% NaCl. Strains Kp20-1R *Paenibacillus* sp., Sa24-1R *Alcaligenes* sp., and Sa25-1S *Myroides* sp. were not resistant to 5% NaCl. Only four isolates grew on a medium with 10% NaCl: strains Se5R, Se1-1R, Se1-3S *Bacillus* sp., and Se3R *Oceanobacillus* sp. Only one strain (Se3R *Oceanobacillus* sp.) was able to grow on a medium with 15% NaCl, but not with 20% NaCl. It should be noted that all the most salt-resistant strains were endophytes of *Salicornia europaea* L.

### 2.6. Properties of Endophytes

The isolates of endophytic bacteria were tested for the presence or absence of cellulase, amylase, protease, and lipase activities, as well as phosphate-solubilizing, nitrogen-fixing ability, and ability to produce IAA (Table 4). All isolates of *Bacillus* sp. belonging to groups 1–6 and 9, as well as the strain Kp20-1R *Paenibacillus* sp. demonstrated cellulase, amylase, and protease activities. Only protease activity was detected for the isolates Se1- 3S (group 7), Sa23S (group 9) *Bacillus* sp., and strain Sa25-1S *Myroides* sp. At the same time, the presence of protease activity was not detected in Se1-1R and Sa21S isolates, also belonging to group 7 *Bacillus* sp. Lipase activity was found only in the strain Kp20-1R *Paenibacillus* sp. 

Of all isolates, only strains Kp20-1R *Paenibacillus* sp. and Kp16R *Pantoea* sp. demonstrated a phosphate-solubilizing ability. The ability to fix nitrogen was found in *Bacillus* sp. isolates belonging to group 1 (with the exception of strain Sa24-3R) and groups 5 and 6, as well as in the strain Kp16R *Pantoea* sp.

As shown in Table 4, almost all strains were able for the secretion of IAA. Strains Kp20-1R *Paenibacillus* sp., Kp16R *Pantoea* sp., and Sa25-1S *Myroides* sp. produced dark red color in the test tubes marked as “++”.

During the growth of isolated endophytes, it was found that the strain Sa24-2R *Bacillus* sp. (group 3) synthesizes a dark brown pigment (presumably melanin) (Figure 3).

### 2.7. Plant Growth Promotion by Endophytes

Figure 4 shows that endophytes had a different effect on the growth of lettuce seedlings. The greatest increase in the fresh weight of seedlings in relation to the control was demonstrated by the strains *Bacillus* sp.: Se1-4S (215%), Bs15S (160%), Kp20-2S (215%), Bs11-2S (85%), Sa21S (265%), Sa23S (125%), Sa25-2S (175%) and the strain Sa25-1S *Myroides* sp. (155%). At the same time, strains Kp19-1R, Kp 19-2R, Se 1-3S *Bacillus* sp., strain Kp20- 1R *Paenibacillus* sp., and strain Kp16R *Pantoea* sp. significantly reduced the weight of lettuce seedlings by 23.4–46.9%.

### 2.8. Antifungal Activities of Endophytes

Antifungal activity of endophytes was tested by the degree of growth suppression of three strains of pathogenic fungi *Fusarium oxysporum* 60519, *Bipolaris sorokiniana* 2716 and *Rhizoctonia solani* 1.2. (Figure 5). All tested strains inhibited the growth of *Bipolaris sorokiniana* 2716 and *Fusarium oxysporum* 60519 with various efficiency (with the exception of the strain Se3R *Oceanobacillus* sp.). On other hand, only 14 strains out of 28 showed antifungal activity against *Rhizoctonia solani* 1.2. The greatest efficacy against all three types of fungi (inhibitory ratio of more than 50%) was found in strains Se1-4S, Kp20-2S, Sa22-1R, Sa24-3R (group 1), Sa24-2R (group 3), Bs11-1R (group 4) and Se5R, Bs11-2S (group 5) *Bacillus* sp.

### 2.9. Polyphasic Characterisation of Endophytes

Figure 6 shows a consensus dendrogram of cluster analysis based on the phenotypic characteristics of strains, indicating the taxonomy, geographical site, host plant, and source of isolation. For clarity, on the dendrogram, we noted the strains with the most pronounced beneficial properties: growth at 55 °C, resistance to 5–15% NaCl, phosphorus-solubilizing activity, high growth stimulation (more than 50% in relation to control), high antifungal activity in relation to three types of fungi (more than 50%). 

We found no strong correlations between growth at 55 °C and taxonomy, host plant, geographical site, and sources of isolation. So, nine strains of *Bacillus* spp. with high-temperature resistance (Se1-4S, Se1-1R, Se1-3S, Se4-1R, Kp19-3S, Se4-2S, Bs12R, Sa22-1R, Bs11-1R) belonged to different taxonomic groups and were isolated from the roots and stems of all four halophytes growing at both sites. However, most of them (five out of nine) were isolated from the *Salicornia europaea* L. 

Most strains demonstrated resistance to 5% NaCl regardless of the geographical site, host plant, and source of isolation. However, all four strains with resistance to 10–15% NaCl (Se1-1R, Se1-3S, Se5R *Bacillus* spp., and Se3R *Oceanobacillus* sp.) were isolated from the roots and stems of *Salicornia europaea* L. growing in the more saline soil in Neftekumsky district. Conversely, all non-salt-resistant strains (Sa25-1S *Myroides* sp., Sa24-1R *Alcaligenes* sp., Kp20-1R *Paenibacillus* sp.) were isolated from plants growing in the significantly less saline soil in Levokumsky district.

Strains with high growth-stimulating activity also belonged to different taxonomic groups and were isolated from all four halophytes, regardless of the geographical site. However, they were all isolated from plant stems, with four of the eight strains being isolated from *Salsola australis* (R.Br.). Both phosphorus-solubilizing strains Kp16R *Pantoea* sp. and Kp20-1R *Paenibacillus* sp. were isolated from *Kochia prostrata* (L.) Schrad. No correlations were found between the high antifungal activity of strains and taxonomy, host plants, geographical sites, and sources of isolation.

## 3. Discussion

The utilization of endophytes for the control of abiotic stresses such as salinity and draught is a relatively unexplored area of research. Endophytes have been studied for over three decades [13,36,37], however, our understanding of their role in plant resistance to salinity and draught is still limited [38,39]. Isolation, identification, and the study of endophytes from halophytic plants could be essential for the development of bioinoculants to facilitate salt soil phytoremediation and be beneficial for mitigating the stress to the plants growing in such salt-affected habitats.

Here we studied endophytes isolated from the four halophytic plants: *Salicornia europaea* L., *Salsola australis* (R.Br.), *Bassia sedoides* (Pall.) and *Kochia prostrata* (L.) Schrad., growing in saline and arid regions of the Stavropol Territory, Russia. It was not possible to isolate endophytes from some plants. For example, only one strain of endophytic bacteria was isolated from the roots of one of five samples of *Salsola australis* (R.Br.), collected in the Neftekumsky district. Perhaps this was caused by the fact that we intentionally used a rather “aggressive” protocol for the surface sterilization of plant material. On the one hand, this approach undoubtedly reduced the number of isolated bacterial strains. On the other hand, the use of this protocol allowed us to be sure that the isolated bacteria are indeed endophytes, and not epiphytes.

It is interesting to note that from the *Salsola australis* (R.Br.), collected in the Neftekumsky district, growing on solonchak soil we isolated only one strain from the roots. While from the *Salsola australis* (R.Br.), collected in the Levokumsky district, growing on kastanozem we isolated nine strains of endophytes. We suppose that soil as well may be a key factor and source of endophytes for the host plant. We isolated eight strains from the *Salicornia europaea* L. This halophyte is the most studied from the point of view of the endophytic microbiome [26,27,28]. Four strains of endophytes were isolated from *Bassia sedoides* (Pall.) and six from *Kochia prostrata* (L.) Schrad. So far, we were the first that isolated endophytes from the plant *Kochia prostrata* (L.) Schrad.

We isolated a total of 28 strains of endophytic bacteria which were identified as belonging to the genera *Bacillus*, *Oceanobacillus*, *Paenibacillus*, *Pantoea*, *Alcaligenes*, and *My-roides*. The vast majority of them—25 strains belonged to the phylum Bacillota (synonym Firmicutes). There were a number of reports about the isolation of endophytes belonging to the phylum Bacillota from halophytic plants [1,26,40]. For example, the endophytes from the leaves of the four dominant halophytes of coastal Gujarat mainly belonged to the *Bacillus* sp., *Paenibacillus* sp. and *Oceanobacillus picturae* [41]. It was reported about the isolation of new bacterial species *Oceanobacillus endoradicis* endophyte from the root of *Paris polyphylla* Smith var. *yunnanensis* [42]. The genera *Pantoea*, *Alcaligenes*, and *Myroides* are also quite common among endophytic bacteria. For example, the halotolerant endophytic bacteria *Pantoea* sp. were isolated from the *Psoralea corylifolia* L., a luxuriantly growing annual weed in salinity-affected semi-arid regions of western Maharashtra, India [43]. It was reported about the isolation of *Myroides odoratimimus* endophyte from *Dactylis glomerata* L. [44].

In our study, nine isolates of *Bacillus* spp. were capable of growth at 55 °C, and two of them—Se1-1R and Se1-3S were also able to grow in 10% of NaCl. It is interesting to note that all these strains were isolated from the *Salicornia europaea* L. These strains can be promising candidates for the development of bioinoculants to facilitate salt soil phytoremediation and be beneficial for mitigating salt stress. Zhao et al. [26] also isolated from the *Salicornia europaea* L. three plant growth-promoting endophytes (*Bacillus endophyticus*, *Bacillus atrophaeus*, and *Bacillus tequilensis*), which demonstrated growth in the liquid medium with 0.51–1.03 M NaCl, this corresponded to 2.5–5.0% NaCl and is comparable with our data. It was reported that of the 20 endophytic bacteria, isolated from the leaves of *Suaeda nudiflora* Moq., *Cressa cretica* L., *Salicornia brachiata* Miq. and *Sphaeranthus indicus* L., seventeen (85%) isolates showed growth at 7.5% NaCl and fifteen (75%) tolerated up to 10% NaCl concentration. The strains *Bacillus foraminis* and *Bacillus gibsonii* could tolerate up to 7.5% NaCl while *Paenibacillus xylanisolvens* tolerated only up to 2.5% NaCl concentration [38]. This was similar to our results.

Most of our endophytes have demonstrated cellulase, amylase, protease activities, nitrogen-fixing ability, and production of IAA. Strain Kp20-1R *Paenibacillus* sp. additionally demonstrated lipase activity. These properties will be helpful in establishing effective plant-microbial symbioses to overcome biotic and abiotic stresses such as phytophatogenic fungi. For example, the enzymes lipase, β-1,3-glucanase, and protease can take part in the degradation of the cell wall of fungi [45,46]. Nitrogen fixation by salt-tolerant bacteria associated with the roots of halophytes is an important source of available nitrogen in saline soils [21,47].

Phosphorus is one of the major essential macronutrients for plants. However, only 0.1% of the total phosphorus is available to plants because of poor solubility and its fixation in soil [48]. Phosphate-solubilizing halotolerant endophytes provide an opportunity to enhance phosphorus availability to plants without exacerbating soil salinity levels [21]. In our study, the ability to dissolve phosphates was revealed in strains Kp20-1R *Paenibacillus* sp. and Kp16R *Pantoea* sp. Both strains were isolated from the *Kochia prostrata* (L.) Schrad. These strains can be used for improving plant growth in phosphorus-deficient saline soils.

All strains except Sa24-1R *Alcaligenes* sp. were capable of producing IAA. IAA has been reported to enhance the protection of bacterial cells against abiotic stresses such as high salt concentrations [49]. Plants can respond to exogenous phytohormones, and these can relieve the adverse effects of salinity [50]. The exogenous application of phytohormones and their precursors provides an attractive approach to counter salt stress conditions by changing the balance of endogenous levels of hormones [51]. IAA can help maintain leaf growth, which helps prevent salinity-induced limitations in plant productivity [52]. However, too high concentrations of IAA can negatively affect plant growth [53,54]. 

One of the strains of *Bacillus* sp.—Sa24-2R synthesized a brown pigment, which possibly may be melanin. Melanin can be synthesized by many soil microorganisms, including bacteria that enter into plant-microbial symbioses. Melanin may play a protective role (at increased temperature, UV radiation, etc.), thus being an additional survival factor under unfavorable conditions.) [55,56]. In addition, it has been shown that purified melanin from endophytic bacteria can exhibit antimicrobial activity. Thus, melanin of strain U7 *Pseudomonas balearica* (the endophyte of the seaweed *Ulva lactuca* L.) was able to inhibit the growth of bacterial phytopathogens *Erwinia chrysanthemi* and *Erwinia carotovora* [57]. Melanin of strain 4NP-BL *Bacillus subtilis* (endophyte from *Salicornia brachiata* Miq.) was active against pathogens *Xanthomonas campestris* and *Alteromonas macleodii* [58].

However, the role of bacterial melanin in direct interaction with the host plant is not clear enough. For example, it was shown that a mutation in the gene encoding tyrosinase (one of the key enzymes of melanin biosynthesis) in nodule bacteria *Sinorhizobium meliloti* did not affect their ability to enter into nitrogen-fixing symbiosis with alfalfa [59]. Therefore, it was concluded that for *S. meliloti* melanin synthesis can be important only in the saprophytic state at the stages of adaptation of free-living cells to the environment. However, opposite results were obtained for the mutant strain CE3 *Rhizobium etli*. The loss of the ability to synthesize melanin led to a decrease in the number and rate of nodule formation on the roots of the host plant [60]. The parent strain CE3 showed a high level of resistance to H_2_O_2_ and phenolic compounds. Similar resistance was found for the melanin-producing phytopathogen *Ralstonia solanacearum* [61]. It has been suggested that tyrosinase may play a role in the early stages of plant-microbial interaction, providing resistance to reactive oxygen species and phenolic compounds formed during plant defense reactions [60,61].

Strains Se1-4S, Ba15s, Kp20-2S, Bs11-2S, Sa21S, Sa23S, Sa25-2S *Bacillus* spp. and Sa25-1S *Myroides* sp. exhibited high growth-promotion activity (85–265% relative to control). It is interesting to note that all of them were isolated from plant stems. Eight strains from the genus *Bacillus* have demonstrated a high degree of growth suppression (59.1–81.2%) of three strains of pathogenic fungi *Fusarium oxysporum* 60519, *Bipolaris sorokiniana* 2716, and *Rhizoctonia solani* 1.2.

The polyphasic analysis did not show strict correlations between the phenotypic characteristics of the strains and their taxonomy, host plant, and geographical site. However, a number of interesting facts were revealed. Thus, most strains with resistance to high temperatures were isolated from *Salicornia europaea* L. (five of the nine strains). Also, all four strains with resistance to 10–15% NaCl were also isolated from the roots and stems of *Salicornia europaea* L. growing in the more saline soil in the Neftekumsky district. Such an effect was demonstrated by other researchers. For example, strains of *Arthrobacter agilis*, *Bacillus endophyticus*, *Bacillus tequilensis*, *Planococcus rifietoensis*, *Variovorax* paradoxus, and ISE-12 *Pseudomonas* sp., isolated from the halophyte *Salicornia europaea* L. were able to facilitate seed germination and seedling growth in the presence of growth inhibitory levels of salt [26,27]. A similar effect was demonstrated by the strain UAE 1 *Micromonospora chelsea* isolated from *Salicornia bigelovii* Torr. [28]. It can be assumed that *Salicornia* as a host plant attracts endophytes with the highest resistance to NaCl. 

In addition, it was found that all strains with high growth-stimulating activity were isolated from the stems of the studied plants. Most often, endophytic and rhizospheric bacteria enter the plant through root hairs using active or passive mechanisms [62]. But only a limited part of the root endophytes is able to penetrate the endoderm and colonize other plant compartments using xylem vessels [63,64]. For further colonization, bacteria need specific penetration mechanisms, as well as the ability to overcome the plant immune response. Therefore, such bacteria can carry special genetic and metabolic determinants that are more suitable for an endophytic lifestyle [65]. It is also possible that at the stage of colonization, the plant is able to select endophytic strains that have certain useful properties, including growth stimulation.

Analyzing all beneficial properties of the tested endophytes we selected *Bacillus* spp. strains Se1-1R, and Se1-3S, capable of growth at 55 °C and 10% NaCl, and strains Sa1-4S, Kp20-2S and Bs11-2S, showing high plant growth promoting as well as antifungal activity, which can be promising candidates for the development of bioinoculants to facilitate salt soil phytoremediation and be beneficial for mitigating the salt stress to the plants growing in salt-affected habitat. This will open up new opportunities in the effective use of cultivated forms of growth-promoting bacteria isolated from the endosphere of halophytic plants to combat the abiotic and biotic stresses of plants and enhance the salt resistance of crops.

## 4. Materials and Methods

### 4.1. Plant Sampling

Halophytic plants were collected in two places in Stavropol Territory, Russia. The first place was near the village of Winter Rate Neftekumsky district (44.5453 N 45.1907 E). The second place was near the village of Burgun-Majary (Kurgan) Levokumsky district (44.5017 N 44.2901 E). Samples were taken at five points in each place. At each point, several halophytic plants were taken together with roots and adjacent soil. Each plant was aseptically placed in a separate paper craft bag and immediately delivered to the laboratory. Plants were identified by the botanist Dr. Nina G. Lapenko and later the identification was checked in the herbarium of the Stavropol Botanical Garden. Herbarium vouchers of collected plants are deposited in North Caucasus Federal Scientific Agrarian Center. 

### 4.2. Surface Sterilization of Plants and Isolation of Endophytic Bacteria

The plant stems and roots were surface sterilized according to [35] with our modifications. The plant samples were washed with tap water, and stems and roots were separated and placed in sterile flasks with 70% ethanol solution (roots for 10 min, stems for 15 min). Then the ethanol solution was drained and the samples were placed in an 18% hydrogen peroxide solution (roots for 15 min, stems for 20 min). After treatment, the samples were washed with sterile water 4 times for three minutes and, then last washed for 30 min.

The sterility check consisted of aliquots of water from the last washing that was plated on GMF (beef enzymatic hydrolysate—15g, NaCl—9g, Agar—15g, NICF, Russia) plates [33]. Plates were incubated at 28 °C for 2 days. The quality of surface sterilization was assessed by the absence of microbial growth on the nutrient medium. Isolation of endophytes was done according to the following protocol. Plant tissues (root, or stem) were aseptically crumbled into smaller fragments using a sterile scissor and then homogenized using a sterile mortar and pestle with 5.0 mL of sterile 0.85% (w/v) NaCl solution. Then, aliquots (100 µL) of serial dilutions (10^−1^–10^−5^) were inoculated onto GMF plates. The plates were maintained at 28 °C for 5 days. A representative of each colony as evident from their colony morphology was picked and transferred to fresh GMF agar plates to establish pure cultures of bacteria. For storage, the cultures were inoculated in agar slant tubes or frozen at −80 °C in 20% glycerol solution. 

### 4.3. Measurement of Salt Content in the Soil

The salt content in the soil was measured according to [66]. The soil in the amount of 10 g was filled with 100 mL of distilled water and centrifuged at 5000 rpm for 5 min. The resulting solution was diluted twice more and measured on a Hanna 3 (Hanna Instruments, Woonsocket, RI, USA) conductometer. Total Dissolved Solids (TDS) can be estimated from the electrical conductivity using the following equation:TDS (ppm) = 0.64 × EC (in μS/cm) = 640 × EC (in dS/m).

### 4.4. Molecular Identification of Endophytes

Total DNA from endophyte strains was isolated with Monarch^®^ Genomic DNA Purification Kit (New England BioLabs^®^ Inc., Ipswich, MA, USA) according to the producer’s instructions. Amplification of DNA fragments was performed by using C1000 TouchTM Thermal Cycler (Bio-Rad, Hercules, CA, USA) and universal primers 27f (5′-AGA GTT TGA TCM TGG CTC AG-3′) and 1525r (5′-AAG GAG GTG WTC CAR CC-3′) [67]. PCR was performed in 30 µL reaction mixtures containing 10× Taq Turbo buffer (Evrogen, Moscow, Russia), 150 µM dNTPs (Evrogen, Moscow, Russia), 5 pM of each primer, 2 U of Taq polymerase (Evrogen, Moscow, Russia), and 10–20 ng of template DNA. PCR parameters were: (1) initial DNA denaturation—95 °C, 3 min; (2) 30 cycles: denaturation—94 °C, 30 s; primer annealing—54 °C, 30 s; elongation—72 °C, 30 s; and (3) final elongation—72 °C, 5 min. 

The obtained PCR fragments were isolated from 1% agarose gel using Monarch^®^ DNA Gel Extraction Kit (New England BioLabs^®^ Inc., Ipswich, MA, USA) and sequenced using an ABI PRISM 3500xl capillary electrophoresis sequencing station (Applied Biosystems, Waltham, MA, USA) according to the manufacturer’s protocol. The strains were identified by comparing the obtained nucleotide sequences of the 16S rRNA gene with the RDP (https://rdp.cme.msu.edu (accessed on 13 September 2022) and GenBank (https://blast.ncbi.nlm.nih.gov (accessed on 13 September 2022) databases. The 16S rRNA gene sequences of all the isolates have been deposited in the GenBank database with the accession numbers OP391268–OP391295. Multiple sequence alignment was conducted by using the Clustal X 1.8 software package. Phylogenetic trees were generated by the Neighbor-Joining method using the MEGA 6.0 software package, and confidence was tested by bootstrap analysis with 500 repeats.

### 4.5. The Effect of Temperature on the Growth of Endophytes

The effect of temperature on the growth of endophytes was studied on GMF medium, Petri dishes were incubated at temperatures of 4; 10; 15; 20; 28; 37; 45; 55, and 60 °C. The intensity of the growth of microorganisms was observed visually after 24 h.

### 4.6. The Effect of Different Concentrations of NaCl on the Growth of Endophytes

The effect of NaCl on the growth of microorganisms was studied on a GMF medium with the addition of various concentrations of NaCl (1; 5; 10; 15; 20%) [68]. Petri dishes were incubated at a temperature of 28 °C. The growth of microorganisms was noted visually after 72 h.

### 4.7. Nitrogen Fixing Ability

To determine the bacteria’s ability to fix molecular nitrogen, an Ashby medium was used (g/L: mannitol or sucrose—20.0, K_2_HP0_4_—0.2, MgSO_4_·7H_2_O—0.2, NaCl—0.2, K_2_SO_4_—0.2, CaCO_3_—5.0, agar—18.0). The nutrient medium was poured into Petri dishes and tested strains were spread on Petri dish by the loop. The Petri dishes were incubated at a temperature of 28 °C for 48 h. Nitrogen-fixing ability was assessed by the presence of bacterial growth. 

### 4.8. Estimation of Phosphorus Solubilization

The ability of isolated endophytic bacteria to dissolve calcium orthophosphates was estimated on the medium of Muromtsev (g/L: glucose—10; asparagine—1; K_2_SO_4—_0.2; MgSO_4_·7H_2_O—0.2; agar—20; yeast extract—0.02; tap water, pH 6.8) according to [69]. Calcium phosphates Ca_3_(PO_4_)_2_ were introduced into the nutrient medium by precipitation. To do this, 3.4 g of CaCl_2_ and Na_3_PO_4_ were added to the sterile molten agar medium (based on one liter) as they dissolved. 1.5 g of freshly formed Ca_3_(PO_4_)_2_ was formed per one liter of medium. The nutrient medium was poured into Petri dishes and tested strains were spread on Petri dishes. The Petri dishes were incubated at a temperature of 28 °C for 120–216 h. The phosphorus-solubilizing activity was judged as the appearance of clear zones around the growth area of a bacterial sample spotted on the plate. 

### 4.9. Estimation of Cellulase Activity

Production of the cellulase, was judged as the appearance of clear zones around the growth of a bacterium on the following solid media according to [33]. Cellulase activity was tested on Getchinson (g/L: NaNO_3_—2.5; K_2_HPO_4_—1.0; MgSO_4_·7H_2_O—0.3; NaCl—0.1; CaCl_2_—0.1; FeCl_3—_0.01; yeast extract—0.1; microcrystalline cellulose—5.0; agar—18.0) agar plates. The Petri dishes were incubated at a temperature of 28 °C for 48 h. Estimation of the clear zones of microcellulose was determined by adding a solution of potassium iodide to a Petri dish. 

### 4.10. Estimation of Amylase Activity

To study the amylase activity, a medium of the following composition was used (g/L: peptone—10; starch—2.0; KH_2_PO_4_—0.3; MgSO_4_·7H_2_O—0.1; agar—15; H_2_O—1000) according to [69]. The strains were inoculated into the center of the Petri dish by injection and Petri dishes were incubated for 5 days at a temperature of 28 °C. Determination of starch absorption zones was determined by adding a solution of potassium iodide to a Petri dish.

### 4.11. Estimation of Protease Activity

To study the protease activity, a medium was used (g/L: casein—20; glucose—0.5; starch—0.5; K_2_HPO_4_—0.3; MgSO_4_·7H_2_O—0.1; agar—15, H_2_O—1000) according to [69]. Tested strains were inoculated into the center of a Petri dish by injection and the activity of casein dissolution was determined after 5 days of cultivation at a temperature of 28 °C. Hydrolysis of casein was detected after treatment of an agar plate with a 10% solution of trichloroacetic acid. The protease activity of the strain was assessed by the hydrolysis zones of casein or their absence.

### 4.12. Estimation of Lipase Activity

To study the lipase activity, a synthetic Seliber medium was used (g/L: K_2_HPO_4_—1; MgSO_4_·7H_2_O—0.3; CaCl_2_—0.3; (NH_4_)_2_HPO_4_—1.5; NaCl—0.3; agar—15; 1 mL of 1.6% aqueous solution of bromothymol blue; H_2_O—1000) according to [69]. 100 µL of sterile sunflower oil was distributed over the surface of the medium in a thin layer. The tested strains were inoculated into the center of a Petri dish by injection. After 5 days of cultivation at a temperature of 28 °C hydrolytic digestion of the oil was noted by a change in the color of the indicator from blue to yellow. The lipase activity of bacterial strains was assessed as the hydrolysis of fats with the formation of fatty acids.

### 4.13. Determination of Secretory Auxin 

The production of auxin (IAA, indole-3-acetic acid) was determined as described by Kamilova and colleagues [70] using the Salkowski reagent [71]. Bacteria were grown in 100 mL R2A medium (Thermo Fisher Scientific, Waltham, MA, USA) with 500 mg/L tryptophane for 72 h at 28 °C at 180 rpm. Bacterial cells were removed by centrifugation for 15 min at 5000 rpm. and the supernatant fluid was subsequently filtered through a 0.22 µm. The reaction was carried out in the ratio of the supernatant/Salkowski reagent—1/2, and the volume of the reaction mixture—1.5 mL. The test tubes were then placed in the dark for 30 min, before the measurement of the OD530 value. A standard curve of IAA content (0.5; 1.0; 5.0; 50.0 µg/mL) was calculated according to absorbance on the spectrophotometer (LEKI, model SS2109UV, Lempäälä, Finland), which was used to estimate IAA concentrations in samples [72].

### 4.14. Estimation of Plant Growth Promotion Ability of Endophytes

The plant growth promotion ability of endophytes was estimated according to [69]. Briefly, the tested strains were grown on a medium of the following composition, g/L: molasses—25.0; corn extract—12.5; MgSO_4_·7H_2_O—0.2; CaCl2—1.0; MnSO_4_—0.1; brewer’s yeast hydrolysate—1.0; pH—7.2. Cells suspensions of 10^7^ cfu/mL were prepared. Seeds of lettuce cv. “Ducat” were placed in sterile test tubes, filled with cell suspensions for each variant, and soaked for 30 min. The seeds in the control were soaked in sterile water. The seeds were germinated in Petri dishes on wet filter paper (20 seeds per dish) in a growth chamber at 28 °C for 5 days. Each variant was performed in three replications. After incubation, the weight of the seedlings was measured on balance (SartoGosm, Saint-Petersburg, Russia).

### 4.15. Antifungal Activities of Endophytes

The antifungal activity of endophytes was determined as described in [44]. Endophytes and pathogen (*Fusarium oxysporum* 60519, *Bipolaris sorokiniana* 2716, *Rhizoctonia solani* 1.2.) were grown on one Petri dish with potato dextrose agar (PDA, Sigma, Burlington, MA, USA) at 25 °C for 7 days. Strains of *Fusarium oxysporum* 60519 and *Bipolaris sorokiniana* 2716 isolated from the seeds of winter wheat grown in the Chechen Republic, were kindly provided by Dr. T.Yu. Gagkaeva, and strain *Rhizoctonia solani* 1.2., isolated from the potato tubers grown in the Leningrad region, was kindly provided by Dr. A.V. Khiutti, All-Russian Research Institute of Plant Protection (Saint-Petersburg, Russia). Endophytes were placed at a distance of 25 mm from the pathogens which were inoculated on the center of the plate. Plates without endophytes were used as controls. Antifungal activity was measured and recorded by using the following formula after the pathogenic fungi in the control plate covered the plate. The experiment was carried out in three independent replicates.
Antifungal activity (%) = [(Control group diameter − treatment group diameter)/Control group diameter] × 100%.

### 4.16. Statistical Analysis

Means were compared by One-Way Analysis of Variance (ANOVA). Significance was computed at *p* < 0.05. The average values of plant growth parameters, and the standard deviation were counted based on several replications.

### 4.17. Polyphasic Analysis

Data on salt resistance, cellulase, protease, lipase activity, phosphorus-solubilizing, nitrogen-fixing ability, and IAA production were presented in the form of a binary matrix (1—presence, 0—absence). For growth at different temperatures, the degree of growth was taken into account as follows: 1—normal growth, 0.75—average growth, 0.5—weak growth, 0—absence. Data on plant growth-promoting activity were presented as a percentage relative to control, divided by 100. Data on antifungal activity were presented as fractions of a unit. The resulting matrix was subjected to cluster analysis using Euclidean distance and Ward as a clustering method.

## 5. Conclusions

A total of 28 strains of endophytes were isolated from 25 plant samples of four halophyte plants growing in saline and arid regions of the Stavropol Territory (Russia). Most of the isolates (23 out of 28) were identified as *Bacillus* sp. while others belonged to the genera *Oceanobacillus, Paenibacillus, Pantoea, Alcaligenes,* and *Myroides*. Two strains of *Bacillus* sp., isolated from the *Salicornia europaea* L. were capable of growth at 55 °C and in 10% of NaCl. Most of the isolated endophytes demonstrated cellulase, amylase, and protease activities, as well as nitrogen-fixing ability and production of IAA. These properties will be helpful in establishing effective plant-microbial symbioses to overcome biotic and abiotic stresses such as phytophatogenic fungi, drought and salinity. Strains Se1-4S, Kp20- 2S and Bs11-2S *Bacillus* spp. demonstrated strong plant growth-promoting activity: 85–265% over control lettuce plants and a high degree of growth suppression (59.1–81.2%) of three strains of pathogenic fungi *Fusarium oxysporum* 60519, *Bipolaris sorokiniana* 2716 and *Rhizoctonia solani* 1.2. Five selected strains can be promising candidates for the development of bioinoculants to facilitate salt soil phytoremediation and be beneficial for mitigating the salt stress to the plants growing in salt-affected habitats.

## Figures and Tables

**Figure 1 plants-11-02992-f001:**
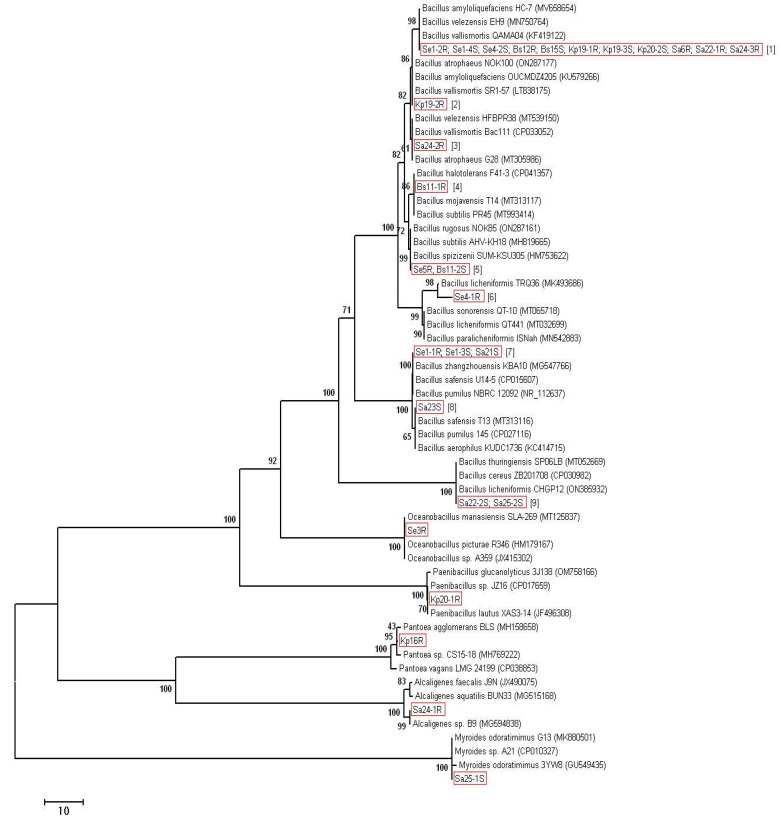
Phylogenetic analysis of 16S rRNA sequences of the bacterial isolates. The Neighbor-Joining clustering method with bootstrap value (500 replicates), pairwise deletion algorithm, and p-distance mathematical model were used. After the name of the isolates, the group number is indicated in square brackets.

**Figure 2 plants-11-02992-f002:**
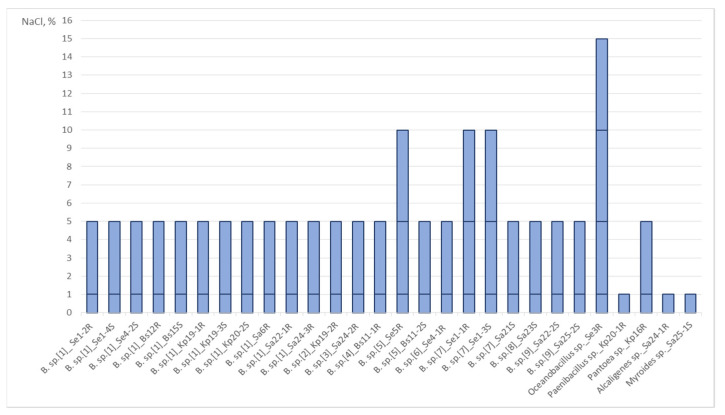
The Effect of Different Concentrations of NaCl on the Growth of Endophytes.

**Figure 3 plants-11-02992-f003:**
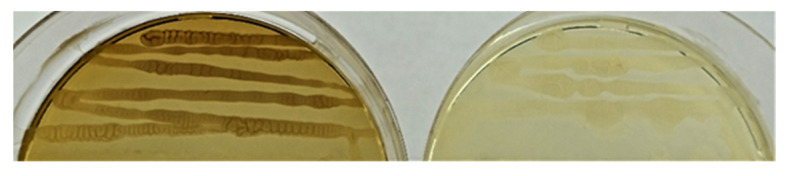
Brown pigment forming strain Sa24-2R *Bacillus* sp. (left); strain Se5R *Bacillus* sp.—does not synthesize pigment (right).

**Figure 4 plants-11-02992-f004:**
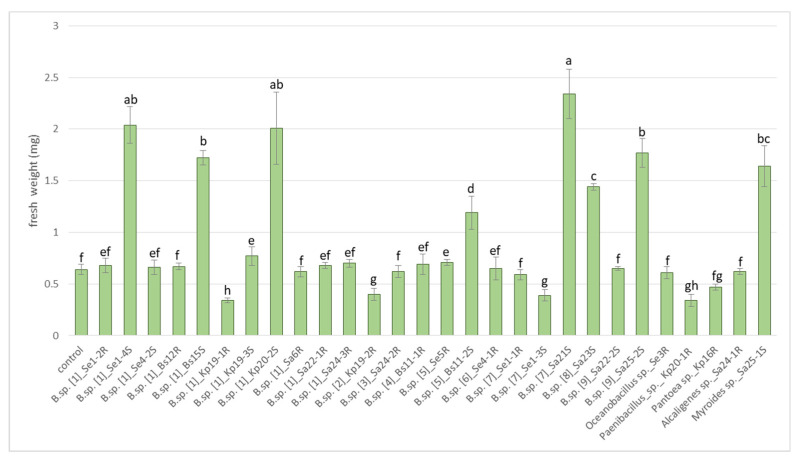
Effect of endophytes on fresh weight of lettuce seedlings cv. “Ducat”. (average of three replicates). Bars represent the mean ± SD of three different experiments, and different letters show a significant difference at the *p* ≤ 0.05 level, as determined by Duncan’s multiple test.

**Figure 5 plants-11-02992-f005:**
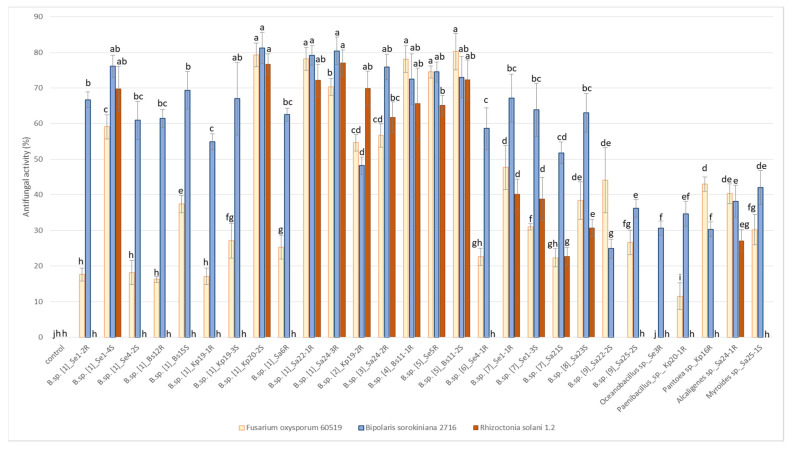
Antifungal activities (%) of endophytes against phytopathogenic fungi *Fusarium oxysporum* 60519, *Bipolaris sorokiniana* 2716, *Rhizoctonia solani* 1.2. Bars represent the mean ± SD of three different experiments, and different letters show a significant difference at the *p* ≤ 0.05 level, as determined by Duncan’s multiple test.

**Figure 6 plants-11-02992-f006:**
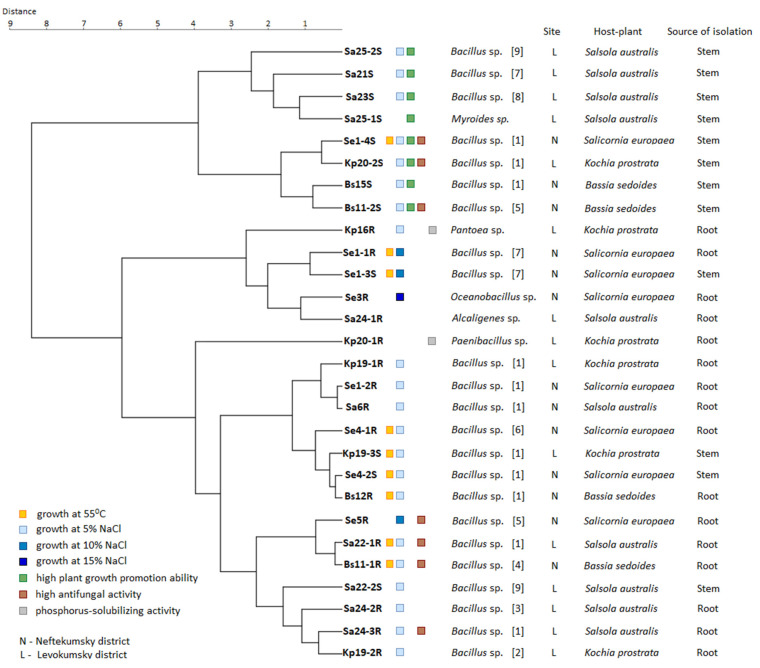
Dendrogram based on phenotypic characteristics of strains showing the relationships between the taxonomy, geographic site, host plant, and source of isolation. The Euclidean distance and Ward clustering methods were used.

**Table 1 plants-11-02992-t001:** Plant samples and isolated strains of endophytes.

Host Plant	Number Plant Sample	Isolation Site	Isolate
Neftekumsky district, Stavropol territory, 44.5453 N 45.1907 E			
*Salicornia europaea* L.	1	Root	Se1-1R; Se1-2R
		Stem	Se1-3S; Se1-4S
	2	-	-
	3	Root	Se3R
	4	Root	Se4-1R
		Stem	Se4-2S
	5	Root	Se5R
*Salsola australis* (R.Br.)	6	Root	Sa6R
	7	-	-
	8	-	-
	9	-	-
	10	-	-
*Bassia sedoides* (Pall.)	11	Root	Bs11-1R
		Stem	Bs11-2S
	12	Root	Bs12R
	13	-	-
	14	-	-
	15	Stem	Bs15S
Levokumsky district, Stavropol Territory, 44.5017 N 44.2901 E			
*Kochia prostrata* (L.) Schrad.	16	Root	Kp16R
	17	-	-
	18	-	-
	19	Root	Kp19-1R; Kp19-2R
		Stem	Kp19-3S
	20	Root	Kp20-1R
		Stem	Kp20-2S
*Salsola australis* (R.Br.)	21	Stem	Sa21S
	22	Root	Sa22-1R
		Stem	Sa22-2S
	23	Stem	Sa23S
	24	Root	Sa24-1R; Sa24-2R; Sa24-3R
	25	Stem	Sa25-1S; Sa25-2S

**Table 2 plants-11-02992-t002:** Molecular identification of endophytes.

Group	Isolate	Identity	Nearest Homolog Sequences (Accession Number)
1	Se1-2R; Se1-4S; Se4-2S; Bs12R; Bs15S; Kp19-1R; Kp19-3S; Kp20-2S; Sa6R; Sa22-1R; Sa24-3R	100%	*Bacillus amyloliquefaciens* strain HC-7 (MW658654) *Bacillus velezensis* strain EH9 (MN750764) *Bacillus vallismortis* strain QAMA04 (KF419122)
2	Kp19-2R	100%	*Bacillus amyloliquefaciens* strain OUCMDZ4205 (KU579266) *Bacillus atrophaeus* strain NOK100 (ON287177) *Bacillus vallismortis* isolate SR1-57 (LT838175)
3	Sa24-2R	100%	*Bacillus atrophaeus* strain G28 (MT305986) *Bacillus vallismortis* strain Bac111 (CP033052) *Bacillus velezensis* strain HFBPR38 (MT539150)
4	Bs11-1R	100%	*Bacillus subtilis strain PR45 (MT993414)* *Bacillus halotolerans strain F41-3 (CP041357)* *Bacillus mojavensis* strain T14 (MT313117)
5	Se5R; Bs11-2S	100%	*Bacillus subtilis* strain AHV-KH18 (MH819665) *Bacillus spizizenii strain SUM-KSU305 (HM753622)* *Bacillus rugosus strain NOK85 (ON287161)*
6	Se4-1R	99%	*Bacillus licheniformis* strain TRQ36 (MK493686) *Bacillus licheniformis* strain QT441 (MT032699) *Bacillus paralicheniformis* strain ISNah (MN542883) *Bacillus sonorensis* strain QT-10 (MT065718)
7	Se1-1R; Se1-3S; Sa21S	100%	*Bacillus pumilus* strain NBRC 12092 (NR_112637) *Bacillus safensis* strain U14-5 (CP015607) *Bacillus zhangzhouensis* strain KBA10 (MG547766)
8	Sa23S	100%	*Bacillus pumilus* strain 145 (CP027116) *Bacillus safensis* strain T13 (MT313116) *Bacillus aerophilus* strain KUDC1736 (KC414715)
9	Sa22-2S; Sa25-2S	100%	*Bacillus thuringiensis* strain SP06LB (MT052669) *Bacillus licheniformis* strain CHGP12 (ON385932) *Bacillus cereus* strain ZB201708 (CP030982)
-	Se3R	100%	*Oceanobacillus manasiensis* strain SLA-269 (MT125837) *Oceanobacillus picturae* strain R346(HM179167) *Oceanobacillus* sp. A359 (JX415302)
-	Kp20-1R	100% 100% 99%	*Paenibacillus lautus* strain XAS3-14 (JF496308) *Paenibacillus* sp. strain JZ16 (CP017659) *Paenibacillus glucanolyticus* strain 3J138 (OM758166)
-	Kp16R	99%	*Pantoea agglomerans* strain BLS (MH158658) *Pantoea* sp. strain CS15-18 (MH769222) *Pantoea vagans* strain LMG 24199 (CP038853)
-	Sa24-1R	100% 99% 99%	*Alcaligenes* sp. strain B9 (MG594838) *Alcaligenes aquatilis* strain BUN33 (MG515168) *Alcaligenes faecalis* strain J9N (JX490075)
-	Sa25-1S	100% 100% 99%	*Myroides odoratimimus* strain G13 (MK880501) *Myroides* sp. strain A21 (CP010327) *Myroides odoratimimus* strain 3YW8 (GU549435)

**Table 3 plants-11-02992-t003:** The Effect of Temperature on the Growth of Endophytes.

Genus	Group	Strain	10 °C	15 °C	20 °C	28 °C	37 °C	45 °C	55 °C
*Bacillus* sp.	[1]	Se1-2R	-	-	+++	+++	+++	+++	-
*Bacillus* sp.	[1]	Se1-4S	-	-	+++	+++	+++	++	++
*Bacillus* sp.	[1]	Se4-2S	-	++	+++	+++	+++	+++	+
*Bacillus* sp.	[1]	Bs12R	-	+	+++	+++	+++	+++	+
*Bacillus* sp.	[1]	Bs15S	-	++	+++	+++	+++	+++	-
*Bacillus* sp.	[1]	Kp19-1R	-	+	+++	+++	+++	+++	-
*Bacillus* sp.	[1]	Kp19-3S	-	++	+++	+++	+++	+++	++
*Bacillus* sp.	[1]	Kp20-2S	-	-	+++	+++	+++	++	-
*Bacillus* sp.	[1]	Sa6R	-	-	+++	+++	+++	+++	-
*Bacillus* sp.	[1]	Sa22-1R	-	+	+++	+++	+++	+++	+
*Bacillus* sp.	[1]	Sa24-3R	-	-	+++	+++	+++	+++	-
*Bacillus* sp.	[2]	Kp19-2R	-	++	+++	+++	+++	+++	-
*Bacillus* sp.	[3]	Sa24-2R	-	++	+++	+++	+++	+++	-
*Bacillus* sp.	[4]	Bs11-1R	-	+	+++	+++	+++	++	+
*Bacillus* sp.	[5]	Se5R	-	++	+++	+++	+++	+	-
*Bacillus* sp.	[5]	Bs11-2S	-	++	+++	+++	+++	+++	-
*Bacillus* sp.	[6]	Se4-1R	-	-	+++	+++	+++	++	++
*Bacillus* sp.	[7]	Se1-1R	-	++	+++	+++	+++	+++	+
*Bacillus* sp.	[7]	Se1-3S	+	++	+++	+++	+++	++	+
*Bacillus* sp.	[7]	Sa21S	-	-	+++	+++	++	++	-
*Bacillus* sp.	[8]	Sa23S	-	-	+++	+++	++	++	-
*Bacillus* sp.	[9]	Sa22-2S	+	+++	+++	+++	+++	+	-
*Bacillus* sp.	[9]	Sa25-2S	+	+++	+++	+++	++	-	-
*Oceanobacillus* sp.		Se3R	-	+	+++	+++	+++	-	-
*Paenibacillus* sp.		Kp20-1R	-	-	+++	++	++	+	-
*Pantoea* sp.		Kp16R	+	++	+++	+++	+++	-	-
*Alcaligenes* sp.		Sa24-1R	-	++	+++	+++	++	-	-
*Myroides* sp.		Sa25-1S	-	+	+++	+++	-	-	-

+++ active growth, ++ medium growth, + weak growth.

**Table 4 plants-11-02992-t004:** Properties of Endophytes.

Genus	Group	Strain	CA	AA	PA	LA	PSA	NF	IAA
*Bacillus* sp.	[1]	Se1-2R	+	+	+	-	-	+	+
*Bacillus* sp.	[1]	Se1-4S	+	+	+	-	-	+	+
*Bacillus* sp.	[1]	Se4-2S	+	+	+	-	-	+	+
*Bacillus* sp.	[1]	Bs12R	+	+	+	-	-	+	+
*Bacillus* sp.	[1]	Bs15S	+	+	+	-	-	+	+
*Bacillus* sp.	[1]	Kp19-1R	+	+	+	-	-	+	+
*Bacillus* sp.	[1]	Kp19-3S	+	+	+	-	-	+	+
*Bacillus* sp.	[1]	Kp20-2S	+	+	+	-	-	+	+
*Bacillus* sp.	[1]	Sa6R	+	+	+	-	-	+	+
*Bacillus* sp.	[1]	Sa22-1R	+	+	+	-	-	+	+
*Bacillus* sp.	[1]	Sa24-3R	+	+	+	-	-	-	+
*Bacillus* sp.	[2]	Kp19-2R	+	+	+	-	-	-	+
*Bacillus* sp.	[3]	Sa24-2R	+	+	+	-	-	-	+
*Bacillus* sp.	[4]	Bs11-1R	+	+	+	-	-	+	+
*Bacillus* sp.	[5]	Se5R	+	+	+	-	-	+	+
*Bacillus* sp.	[5]	Bs11-2S	+	+	+	-	-	+	+
*Bacillus* sp.	[6]	Se4-1R	+	+	+	-	-	+	+
*Bacillus* sp.	[7]	Se1-1R	-	-	-	-	-	-	+
*Bacillus* sp.	[7]	Se1-3S	-	-	+	-	-	-	+
*Bacillus* sp.	[7]	Sa21S	-	-	-	-	-	-	+
*Bacillus* sp.	[8]	Sa23S	-	-	+	-	-	-	+
*Bacillus* sp.	[9]	Sa22-2S	+	+	+	-	-	-	+
*Bacillus* sp.	[9]	Sa25-2S	+	+	+	-	-	-	+
*Oceanobacillus* sp.		Se3R	-	-	-	-	-	-	+
*Paenibacillus* sp.		Kp20-1R	+	+	+	+	+	-	++
*Pantoea* sp.		Kp16R	-	-	-	-	+	+	++
*Alcaligenes* sp.		Sa24-1R	-	-	-	-	-	-	-
*Myroides* sp.		Sa25-1S	-	-	+	-	-	-	++

Legend: CA—cellulase activity, AA—amylase activity, PA—protease activity, LA—lipase activity, PSA—phosphorus-solubilizing activity, NF—nitrogen-fixing activity, IAA—production of IAA; (+)—presence of activity, (-)—absence of activity; For IAA products: (+)—availability of IAA production (pink color sample), (++)—availability of high IAA production (dark red color sample).

## Data Availability

The data presented in this study are available upon request from the corresponding author.

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
