# Peer review of "Endophytes from Halotolerant Plants Aimed to Overcome Salinity and Draught"

_plants, 2022, doi:10.3390/plants11212992_

Round 1

Reviewer 1 Report

In the manuscript, authors isolate and characterize endophytic bacteria from four different plant species characterized by halotolerance and ability to live in “extreme” environments. Authors investigate the beneficial properties of endophytic bacterial strains, that could be helpful for the plants to overcome abiotic stresses.

The topic is really interesting, but the manuscript needs to be improved in many aspects before publishing.

Major concerns and suggestions

1) authors describe the same sterilization protocol for the external parts of the four plant species studied. I’m not sure about the suitability of the same protocol for all the four plant species, that show very different characteristics. Moreover, it seems that the sterilization protocol used is very “aggressive”, and this could explain the very low number of isolated strains in the plant material. Could author provide references about the used method, and a critical discussion on the possibility that such protocol might have affected the number of isolates bacterial strains?

2) for a better understanding of the very high amount of results, I suggest evaluating to replace Fig.2 and Fig. 4 with a different representation, for example a table, in which the growth of microorganisms is indicated with “+”, and the lack of growth with “-“. Authors might also evaluate intermediate levels of growth, for example with other symbols (i.e “+-“), or by attributing values to the growth respect to a control.

3) In order to better enhance the quality and the amount of produced data, I suggest to the authors to evaluate the possibility of introducing a multivariate analysis that takes into consideration all the produced data related to each strain and to the isolation source (i.e. source of isolation, taxonomy, phenotypic characteristics, salinity of the soil etc.). According to this suggestion, authors will be able to provide not only a description of the abilities and characteristics of each isolated strain, but also to individuate eventual clusterizations of different bacterial groups, on the bases of the parameters (i.e. do they share similar characteristics according to the isolation source? Do they share similar characteristics according to taxonomy or to the plant species?). In my opinion, this might improve the discussion session.

Minor suggestions

P3 L105: shoots: do authors mean “stems”?

P3 L128: was/were: please correct

P9 paragraph 2.7 and Table 2: authors provide results of the IAA production by endophytic isolated strains. Are those values normalized with the number of cells in the liquid culture? Indeed, values are reported as micrograms of IAA produced by 1 ml of liquid culture; but if the number of bacterial cells in the liquid cultures of each bacterium are not the same, or at least comparable (i.e. measured using OD 600), the IAA production results are not comparable as well among each other. Please, provide details.

P14 L425: please check reference 28 because I’m not sure it is the right one to cite here. The same for reference 38 (P16 L534).

P14 L427-432: Can author provide the cell colony enumeration in the serial dilution plates? This datum is important since it gives an idea of the microbial load in each processed sample, and might be an important focus in the discussion of the sterilization protocol used, and previously commented (see previous comments and suggestions).

P14, paragraph 4.2: details about chemicals used in the PCR are missing, as well as the citations of the used protocol.

P15 L 477: brackets must be closed

P15 L 490: “…agar- 18.0). agar plates.” Please correct

P16 paragraph 4.14: the references to the PGP protocol are missing in the paragraph

P16  L 547: please replace 2.16 with 4.16

Reviewer 2 Report

1. The botanist that identified the plants should be stated.

2. Please provide more information about how the plant was aseptically bagged.

3. Line 139, please correct to Bacillus ssp.

4. Please indicate in a Table the temperatures at which the samples were harvested. This information is important when assessing the different bioactivities.

5. Line 495, please correct to "according to."

6. Please correct the typos in the manuscript.

7. In section 4.15., please provide information about the source of the strains and the meaning of the numbers.

8. The legend in Figures 4 and 5 should be completed.

9. In Figure 5, do not use melanin-like because the identity of the pigment is unknown. 

9. In section 2.7., correct the word "prduction."

10. In Table 3, define IAA at the bottom of the Table.

11. Figure 6 is missing a statistical analysis. The legend should be completed as well.

12. The protocol in 4.7 is not clear. How were the endophytes placed on the plate? in what medium? What concentration?

13. Figure 7 is missing controls. Also, the legend should be expanded.

14. In Line 335, please compare the paragraph with the literature. Regarding the potential application of the mentioned enzymes, are these enzymes secreted to the environment?

15. Line 343-367, please compare to the literature.

16. The comments in line 368 are misleading. The authors have not checked for the presence of melanin, and the color of the culture cannot be used assuredly.

17. Line 392, please compare with the literature.

18. The brand of the instruments, including manufacturer and country, should be stated.

Reviewer 3 Report

The manuscript titled “Endophytes from halotolerant plants aimed to overcome salin-2 ity and draught.” reported the endosphere of four halophytic plants: Salicornia europaea L., Salsola australis (R.Br.), Bassia sedoides (Pall.) and Kochia prostrata (L.) Schrad. from arid and saline areas of the Stavropol Territory, Russia. This is a well-written article and I anticipate that the manuscript should be of great interest. Before recommending this article for publication, there are some shortcomings that should be resolved.

General comments

Overall, the study is well designed and presented in a good way. 

Abstract

The authors elaborated the abstract in a good way.

Introduction

Add the following references in lines 34, 51 and 57 accordingly.

10.3390/agronomy11030409 

10.1080/17429145.2022.2091801

10.1007/s13213-019-01470-x

Materials and Methods

Add the relevant reference in line 420.

This section is very lengthy, kindly avoid extra explanation.

Put the relevant references for lines 467, 473, 485, 493, 499, 507, 516, 532.

Results and Discussion

These sections are well written.

Line 133-134: Kindly, re-structure the statement “A collection of endophytes of halophytic plants was formed from 133 the isolated isolates, stored at a temperature of -800 C.”

Author Response

Seea attached file

Round 2

Reviewer 1 Report

Authors significantly improved the revised version of the manuscript, and they also added new analysis as suggested.

Some minor concerns are listed below.

 The references of the used sterilization method must be added in the text, not only in the answer to reviewer, even if the protocol is modified in this version.

A (brief) critical discussion on the choice of the sterilization protocol is missing in discussion session, as well as the possibility that the chosen protocol might have affected the number of isolated strains.

Authors did not integrate the discussion with the new analysis reported in Fig. 6, that in my opinion, is really interesting.

P2 L69:” sinensesignificantly”. Please, divide the two words.

P4 L151: You mention Figure 3 but I think it is Figure 1 (phylogenetic analysis)

P8 L208: produse/produce please correct

Reviewer 2 Report

- The name of the botanist has not been included in the manuscript.

- Comment 4 has not been addressed in the manuscript.

- Comment 8, please add more information to the mentioned figures. It should stand by itself.

- Comment 12, the statistical analysis is missing.

- Figure 7 disappeared??

Round 3

Reviewer 2 Report

The authors have partially responded to my concerns.

The authors did not highlight the changes in the manuscript and it is hard to follow them.

Please provide a new version highlighting the changes.

I can't find the name of the botanist. Highlight in the manuscript.

The Legend in Fig 4 has 2 fonts.
In Figure 4, the meaning of IAA should be defined in words at the bottom.
Typos have not been totally corrected.
Figure 7 disappeared.

Round 4

Reviewer 2 Report

The authors have sent a revised version of the manuscript.

Comment 1. The authors misunderstood my request. Since they have collected plants, the identification of the species should be done by a botanist and a voucher should be deposited in a herbarium. Please complete this information.

Comment 3. The authors should be more explicit about the legends. They wrote Bars show +- SD. It is missing what they show. Probably a better way will be Bars represent the mean +- SD of three different experiments.
